# Intestinal Epithelial Co-Culture Sensitivity to Pro-Inflammatory Stimuli and Polyphenols Is Medium-Independent

**DOI:** 10.3390/ijms25137360

**Published:** 2024-07-04

**Authors:** Michelle J. Haddad, Juanita Zuluaga-Arango, Hugo Mathieu, Nicolas Barbezier, Pauline M. Anton

**Affiliations:** 1Transformations et Agroressources, ULR 7519, Institut Polytechnique UniLaSalle, Université d’Artois, 60000 Beauvais, France; michelle.haddad@unilasalle.fr (M.J.H.); juanita.zuluagaarango@etu.unilasalle.fr (J.Z.-A.); hugo.mathieu@unilasalle.fr (H.M.); nicolas.barbezier@unilasalle.fr (N.B.); 2HCS Pharma, 59120 Loos, France

**Keywords:** intestinal epithelium, in vitro co-culture systems, RPMI-1640, LPS, cytokine stimulation, catechins, inflammatory and apoptotic pathways

## Abstract

The complexification of in vitro models requires the compatibility of cells with the same medium. Since immune cells are the most sensitive to growth conditions, growing intestinal epithelial cells in their usual medium seems to be necessary. This work was aimed at comparing the sensitivity of these epithelial cells to pro-inflammatory stimuli but also to dietary polyphenols in both DMEM and RPMI-1640 media. Co-cultures of Caco-2 and HT29-MTX cells were grown for 21 days in the two media before their stimulation with a cocktail of TNF-α (20 ng/mL), IL-1β (1 ng/mL), and IFN-γ (10 ng/mL) or with LPS (10 ng/mL) from *E. coli* (O111:B4). The role of catechins (15 µM), a dietary polyphenol, was evaluated after its incubation with the cells before their stimulation for 6 h. The RPMI-1640 medium did not alter the intensity of the inflammatory response observed with the cytokines. By contrast, LPS failed to stimulate the co-culture in inserts regardless of the medium used. Lastly, catechins were unable to prevent the pro-inflammatory response observed with the cytokines in the two media. The preservation of the response of this model of intestinal epithelium in RPMI-1640 medium is promising when considering its complexification to evaluate the complex cellular crosstalk leading to intestinal homeostasis.

## 1. Introduction

Intestinal homeostasis is an important contributor of human health since the intestine is in charge not only of transforming and absorbing nutrients, conditioning further metabolic activities, but also of preventing the entrance of noxious substances (such as xenobiotics or neoformed compounds) [1]. To reach such an organization, the microbiota, the host intestinal epithelium, and the immune system need to closely communicate through the release of many mediators in charge of permanently adjusting these dynamic interactions. However, when disrupted, the imbalance between gut commensal microbiota and the host response in the intestinal mucosa leads to inflammatory disorders among which are inflammatory bowel diseases (IBDs). These diseases are of unknown origin, but the scientific community agrees on the fact that they are associated with an imbalance of the gut microbiota and an uncontrolled immune response, both leading to intestinal epithelial disruption [2]. Foods and their components (nutrients and contaminants) are among the causes regularly evoked as contributing to the exacerbation or the limitation of the inflammatory events [3]. The understanding of the causes of these diseases has been under investigation for several years, with the use of multiple approaches ranging from simple culture models to animal approaches and clinical observations. What is happening at the intestinal epithelial level at the origin of the inflammatory response is still not fully understood, because none of the models proposed so far can mimic the exact events taking place in the human intestine.

Given that they display those close interactions between (1) the microbiota, (2) the epithelium, and (3) the immune system, animal models can contribute to a better understanding of the pathophysiological stages of chronic inflammation. Nevertheless, mainly for ethical reasons, understanding the chronology of the induction of the inflammatory response and barrier disruption at the intestinal level cannot be assessed with animal models. Hence, multiple attempts to reproduce in vitro such interactions have been made and described in the scientific literature. Most of them are based on 2D models (cells grown on a flat surface), which may use transformed cell lines or cells coming from resected tissues. More recently, some 3D approaches (cells grown on a three-dimensional gel-like matrix) have been published [4,5], attempting to reproduce intestine–microbiota interactions. However, the close communication between the immune and epithelial cells is still poorly studied for methodological reasons mainly linked to the kinetics of proliferation and cell growth medium compatibility. Among the challenges still to be considered, growth media are of the upmost importance since commercial cell lines have been engineered to fit with a specific one and this is particularly true with immune cell lines which are among the most sensitive ones. One of the most frequently used ones is the THP-1 cell line, which will, when incubated under appropriate conditions in RPMI-1640 medium, differentiate into macrophages [6]. The literature has shown that THP-1 can only grow in RPMI-1640 medium. Therefore, to create a functional model reproducing the intestinal epithelium, the growth medium of the other cell components needs to be adjusted to prevent the alteration of the cellular response. The commonly used cell model for the intestinal epithelium is a co-culture of Caco-2/HT29-MTX (for a review, see [7]). Those two cell types grow and differentiate perfectly into, respectively, enterocyte and goblet cell phenotypes in DMEM medium. Since cells are quite sensitive to the medium in which they are grown, it would be important to assess whether the model of intestinal epithelial cells in RPMI-1640 medium could behave similarly as what is observed in DMEM medium. However, their response to these stimuli in other media such as RPMI-1460, with the purpose of co-cultivating them with immune cells, has never been described. When stimulated with pro-inflammatory cytokines [8] or with lipopolysaccharide (LPS) from *E. coli* [9], the Caco-2/HT29-MTX co-culture in DMEM medium responds by activating pro-inflammatory and pro-apoptotic pathways and by releasing mediators such as interleukin 8 (IL-8) [8]. Furthermore, this cell model has been widely used to evaluate the protective effects of a large number of molecules including functional food ingredients with the purpose of evaluating, in vitro, the putative benefits of these ingredients for preventing or alleviating the gut inflammatory response in healthy or IBD patients. Among functional ingredients, polyphenols are phenolic compounds largely represented in our food [10]. These natural substances extracted from plants possess anti-inflammatory and antioxidant properties that could contribute to the prevention of an inflammatory reaction at the epithelial level [11]. Among their mechanisms of action, their immunomodulatory effect has also been suggested [12]. Among them, catechins are polyphenols mainly found in green tea leaves. They have been widely studied for their antioxidant, antibacterial, and anti-inflammatory properties [13], but so far, their protective effects on Caco-2 cells have been suggested but never clearly demonstrated [14].

According to the literature, so far, no study has tackled the response of this widely used co-culture in media other than DMEM, and so the purposes of this study were to assess (1) whether the change in growth medium from DMEM to RPMI-1640 could modify the response of the Caco-2/HT29-MTX co-culture to pro-inflammatory cytokines and LPS stimuli, and (2) whether the incubation with catechins of the co-culture grown in those media could prevent or limit their response to pro-inflammatory stimuli, as a model of functional food evaluation.

## 2. Results

### 2.1. Time-Dependent Effects of the Pro-Inflammatory Cytokine Cocktail Stimulation on the Caco-2/HT29-MTX Co-Culture Are Medium-Dependent

#### 2.1.1. Effect of Medium on the Kinetics of Pro-Inflammatory Gene Stimulation

In DMEM medium, the incubation of the Caco-2/HT29-MTX cells for 2 and 6 h with the cocktail of cytokines led to a significant (*p* < 0.05) increase in CXCL8 gene expression after 2 h (4.45 ± 0.14) as well as after 6 h (5.72 ± 0.26) (Figure 1a).

In RPMI-1640 medium, the incubation of the Caco-2/HT29-MTX cells for 2 and 6 h with the cocktail of cytokines also resulted in a significantly upregulated (*p* < 0.05) expression of CXCL8 after 2 h (6.03 ± 0.44) as well as after 6 h (4.25 ± 0.49) (Figure 1b).

The incubation of the co-culture with the RPMI-1640 medium does not modify the kinetics of expression of CXCL8 following its stimulation with a cocktail of pro-inflammatory cytokines.

#### 2.1.2. Effect of Medium on the Kinetics of IL-8 Secretion

The incubation of Caco-2/HT29-MTX cells during 2 or 6 h with a cocktail of pro-inflammatory cytokines led to a time-dependent significant (*p* < 0.05) increase in IL-8 secretion after 2 h (389.1 ± 37.9 vs. 136.5 ± 12.4 pg/mL in the unstimulated cell group) and 6 h (866.0 ± 135.5 vs. 136.5 ± 12.4 pg/mL in the unstimulated cell group) of exposure in DMEM medium (Figure 2a).

The incubation of Caco-2/HT29-MTX cells during 2 or 6 h with a cocktail of pro-inflammatory cytokines also resulted in a time-dependent significant (*p* < 0.05) increase in IL-8 secretion after 2 h (269.6 ± 5.3 vs. 116.7 ± 1.6 pg/mL in the unstimulated cell group) and 6 h (772.4 ± 102.4 vs. 116.7 ± 1.6 pg/mL in the unstimulated cell group) of exposure in RPMI-1640 medium (Figure 2b).

The stimulation of the co-culture in 24-well plates by the cocktail of pro-inflammatory cytokines resulted in the upregulation of CXCL8 expression. It was followed by an upregulation of Il-8 secretion regardless of the timepoint considered and growth medium used.

### 2.2. Epithelial Cell Response Is Dependent on the Stimulus but Not on the Medium

#### 2.2.1. Medium-Independent Sensitivity of Epithelial Cell in Response to Pro-Inflammatory Stimuli

Effect on epithelial permeability.

Exposing Caco-2/HT29-MTX cells grown in Transwell^®^ chambers to LPS from *E. coli* or to a cocktail of pro-inflammatory cytokines for 6 h did not alter the intestinal epithelial cell permeability measured by TEER either in DMEM medium (Figure 3a) or in RPMI-1640 medium (Figure 3b).

The exposure of the model of intestinal epithelium in Transwell^®^ chambers to either LPS from *E. coli* or the cocktail of cytokines did not result in any modification of TEER in the two media.

Effect on CXCL8 and NF-κB pro-inflammatory gene expressions.

Exposing h Caco-2/HT29-MTX cells in Transwell^®^ chambers to LPS from *E. coli* for 6 did not stimulate gene expressions of CXCL8 and of NF-κΒ either in DMEM medium (Figure 4a,c) or in RPMI-1640 medium (Figure 4b,d).

By contrast, the stimulation of the co-culture to the cocktail of pro-inflammatory cytokines during 6 h significantly increased (*p* < 0.05) CXCL8 gene expression both in DMEM medium (2.09 ± 0.59) (Figure 4a) and in RPMI-1640 medium (16.61 ± 5.56) (Figure 4b). However, neither the LPS of *E. coli* nor the cocktail of cytokines were able to modulate NF-κB gene expression in DMEM medium or in RPMI-1640 medium (Figure 4c,d).

The stimulation of the epithelium in Transwell^®^ chambers to the cocktail of cytokines resulted in the upregulation of the expression of the CXCL8 gene but not that of NF-κΒ gene in both the DMEM and the RPMI-1640 media.

Effect of medium on IL-8-induced secretion

Exposing Caco-2/HT29-MTX cells in Transwell^®^ chambers to LPS from *E. coli* for 6 h did not stimulate the secretion of IL-8 either in DMEM medium (Figure 5a) or in RPMI-1640 medium (Figure 5b).

By contrast, the exposure of the co-culture to the cocktail of pro-inflammatory cytokines resulted in a significant (*p* < 0.05) release of IL-8 in the apical DMEM medium (566.5 ± 78.7 vs. 20.0 ± 2.9 pg/mL in unstimulated cells) (Figure 5a). However, this increase was not significant in RPMI-1640 medium (343.9 ± 58.7 vs. 28.5 ± 2.6 pg/mL in unstimulated cells) (Figure 5b).

The stimulation of the co-culture in Transwell^®^ chambers only resulted in a significant secretion of IL-8 in DMEM medium.

#### 2.2.2. Medium-Independent Sensitivity of Epithelial Cells Exposed to Polyphenols before Their Stimulation by a Pro-Inflammatory Cytokine

Effect on epithelial permeability.

Exposing Caco-2/HT29-MTX cells grown in Transwell^®^ chambers to the cocktail of pro-inflammatory cytokines for 6 h did not alter the intestinal epithelial cell permeability measured by TEER either in DMEM medium (Figure 6a) or in RPMI-1640 medium (Figure 6b).

Moreover, neither the exposure for 6 h to catechins alone, nor the exposure to catechins with the cocktail of cytokines was able to modify the epithelial permeability in DMEM medium (Figure 6a) or in RPMI-1640 medium (Figure 6b).

Catechins did not change the TEER in the two media, regardless of whether the cells were stimulated or not.

Effect on pro-inflammatory NF-κΒ and CXCL8 gene expressions.

The stimulation of Caco-2/HT29-MTX grown in Transwell^®^ chambers by the cocktail of pro-inflammatory cytokines for 6 h did not result in a significant increase in the expression of CXCL8 or NF-κΒ genes either in the DMEM medium (Figure 7a,c) or in the RPMI-1640 (Figure 7b,d).

Furthermore, the incubation of the co-culture with catechins did not result in any modification of CXCL8 and NF-κΒ gene expressions in either the DMEM medium (Figure 7a,c) or the RPMI-1640 medium (Figure 7b,d).

Lastly, the incubation of the co-culture with catechins before their stimulation with the cocktail of pro-inflammatory cytokines did not modify the expression of CXCL8 and NF-κΒ genes in either the DMEM (Figure 7a,c) or the RPMI-1640 (Figure 7b,d) medium.

Catechins did not prevent the upregulation of CXCL8 observed with the cocktail of cytokines regardless of the medium used.

Effect of medium on IL-8-induced secretion

The exposure of Caco-2/HT29-MTX cells grown in Transwell^®^ chambers to the cocktail of pro-inflammatory cytokines for 6 h in the absence or presence of catechins significantly (*p* < 0.05) stimulated the secretion of IL-8 in DMEM medium (566.5 ± 78.7 vs. 20.0 ± 2.9 pg/mL in unstimulated cells) (Figure 8a) but not in RPMI-1640 medium (343.9 ± 58.7 vs. 28.5 ± 2.6 pg/mL in unstimulated cells) (Figure 8b).

The exposure of the co-culture to catechins alone did not stimulate the secretion of IL-8 either in the apical DMEM medium (Figure 8a) or the apical RPMI-1640 medium (Figure 8b).

We did not observe any reduction in IL-8 secretion in cells pre-incubated with catechins before their stimulation with the cocktail of cytokines in either medium (Figure 8a,b).

The upregulation of IL-8 secretion by the cocktail of cytokines was only slightly limited by catechins in DMEM medium.

Effect on pro-apoptotic pathway CASP3 and CASP9 gene expressions.

The exposure of Caco-2/HT29-MTX cells to the cocktail of pro-inflammatory cytokines did not activate the expression of CASP3 either in the DMEM medium or in the RPMI medium (Figure 9a,b). Furthermore, this stimulation did not upregulate the expression of CASP 9 gene in either medium (Figure 9c,d).

The incubation of the co-culture with catechins did not modify the level of expression of CASP3 or CASP9 in the two media (Figure 9). Lastly, the catechins did not change the level of expression of either CASP 3 or CASP9 in the presence of the cocktail of pro-inflammatory cytokines (Figure 9).

The apoptotic caspase-3 and caspase-9 were not activated by the cocktail of pro-inflammatory cytokines either in DMEM medium or in RPMI-1640 medium, and catechins did not act on these pathways.

## 3. Discussion

The first objective of this study was to compare the response of the Caco-2/HT29-MTX co-culture, as a model of intestinal epithelium, in their conventional medium, DMEM, and in the RPMI-1640 medium, usually dedicated to immune cell culture for the complexation of the cell model. The seeding was operated at a ratio of 9:1 for all the experiments realized in this study. We previously evidenced that this ratio was the most appropriate one to evaluate the effects of molecules on the intestinal epithelium in Transwell^®^ chambers [8].

To answer this question, we first assessed whether the response of the co-culture to the cocktail of pro-inflammatory cytokines [8] could be modified by the culture medium. It is indeed well established that an inappropriate culture medium could be a source of stress for cells and could change their sensitivity to exogenous stimuli or promote spontaneous pro-inflammatory pathways not seen under standard conditions [15]. The first challenge was to determine whether this new cell culture condition would be suitable to work with a fully differentiated epithelium. We observed that the RPMI-1640 medium did not alter either the expression of CXCL8 or the secretion of IL-8 in the apical medium. These results confirm that RMPI-1640 medium did not alter the growth and differentiation of either the Caco-2 or the HT29-MTX cell lines since the amplitude of responses in RPMI-1640 and in DMEM as a control medium were quite similar at both 2 and 6 h. Furthermore, the response observed after stimulating the co-culture with the cocktail of cytokines is consistent with our previous results [8].

Based on this first set of data, we decided to move to the evaluation of the response of cells in Transwell^®^ chambers. This technique is more representative of the physiological exchanges happening at the level of the intestinal epithelium. The second aim of this study was to evaluate the response of the model of intestinal epithelium to LPS from *E. coli* and compare this response to the one observed with the cocktail of cytokines in both DMEM and RPMI-1640 media. Under physiological conditions, the intestinal epithelium is in permanent contact with the microbiota and possesses at its surface pattern recognition receptors (PRRs) such as Toll-like receptors (TLRs) which are able to bind to bacterial toxins and stimulate the pro-inflammatory response. The present study was conducted with a 6 h stimulation by LPS at a concentration of 10 ng/mL [16]. We did not observe any modification of the epithelial permeability with either LPS or the cocktail of cytokines in the two media. By contrast, while LPS failed to stimulate CXCL8 gene expression, the expression of this gene was upregulated in the presence of the cytokines in both DMEM and RPMI-1640 media. This was associated with a clear secretion of IL-8 which was only significant in the DMEM medium. A previous study indicated that 1 µg/mL LPS induced an inflammatory response of Caco-2 cells without causing cell death [17]. Furthermore, the pro-inflammatory effect of LPS at this concentration on Caco-2 cells was significantly limited by *Lactobacillus plantarum* [9]. Another work evidenced that unstimulated Caco-2 cells did not express TLR4, the well-known receptor for *E. coli* LPS, while the expression of the gene and its protein were clearly upregulated by the exposure of Caco-2 to LPS (O26:B6) for 24 h at concentrations ranging from 5 to 50 µg/mL [18]. This was associated with an increase in DNA fragmentation and caspase-3 activation. Those data are contradictory to our results, but the experimental conditions may explain such discrepancies. In contrast with all the previous experiments conducted with this toxin, the stimulus applied here was on a co-culture of Caco-2/HT29-MTX cells grown on Transwell^®^ chambers for 21 days. We cannot exclude here that an incubation of the co-culture with LPS for 24 h could have resulted in a stimulation of pro-inflammatory pathways. However, the Transwell^®^ Caco-2 model is not responding properly to pro-inflammatory stimuli for such a length of time because after a 24 h nutrient deprivation (i.e., in a serum free medium), Caco-2 cell intracellular activity is altered and transepithelial permeability is increased [19]. Another explanation could be that published studies were run most of the time on Caco2 cells grown for 48–72 h [9] and not always on Transwell^®^ chambers [18]. At this timepoint of cell culture, the epithelium is not fully functional, cell junctions are not established [20], and cells may be more vulnerable to noxious stimuli. Furthermore, with this model of co-culture on Transwell^®^ chambers, the secretion of mucus by HT29-MTX may have partly shielded the Caco-2 cells. They may have prevented the anchorage of LPS on TLR4 receptors and consequently the activation of pro-inflammatory pathway signaling at the origin of the upregulation of CXCL8 gene expression and the secretion of the chemokine. Lastly, in our study, we worked with the O111:B4 serotype of LPS from *E. coli.* In other studies, authors applied on a monoculture of Caco-2 cells, grown for 72 h in DMEM medium, the O127:B8 or the O26:B6 serotypes of LPS from *E. coli* [21,22]. The difference in experimental conditions could modify the reactivity of the cells. In other words, the response of Caco-2 cells seems to be tightly linked to (a) the environment in which they are grown, i.e., simple wells vs. Transwell^®^ systems, (b) the fact of being coupled or not with a goblet cell line, (c) the growth length (in days), and (d) the concentration and duration of the stimulation with LPS but not of the medium. It is noteworthy that our stimulatory conditions were the lowest in terms of concentration and duration. They were realized on fully differentiated cells. Altogether, these conditions certainly contribute to the weak answer observed. Between all the stimulating conditions used, ours get closer to in vivo conditions. Those results are of peculiar interest when considering multiple cultures with immune cells such as THP-1 differentiated in macrophages. Since LPS is able to stimulate macrophage activation and their subsequent release of pro-inflammatory signals such as TNF-α or IL-1β [21], its application on a co-culture of macrophages and epithelial cells could activate the pro-inflammatory response of the epithelial cells as observed in vivo. Such an approach was recently conducted on a 3D flipwell model using *B. subtilis* [23].

This set of experiments also provided evidence on the suitability of RPMI-1640 to grow the co-culture although the upregulation of the CXCL8 gene and the secretion of the protein in the apical medium was lower than in their usual DMEM medium. Both the DMEM and the RPMI-1640 medium contain a balanced mixture of essential nutrients including amino acids and vitamins and a sodium bicarbonate buffer to help to maintain a physiological pH at a 5–10% CO_2_ environment. None of them contain growth factors which were brought by the same FBS. However, the main difference between the two media comes from the fact that RPMI-1640 contains glutathione (a reducing agent) and high concentrations of vitamins such as biotin, vitamin B12, and parabenzoic acid (PABA). Such compounds contribute to oxidative stress protection and prevent the impairment of DNA synthesis and methylation which are of prime importance for adequate cell growth [24,25].

The absence of upregulation of NF-κΒ under those conditions is intriguing. However, it could be due to the final timepoint chosen (6 h). The literature clearly indicates that TNF-α is responsible for a quick activation of this transcription factor at the origin of the upregulation of the promoter of the CXCL8 gene which is fully consistent with our data [26]. However, we did not observe any change in NF-κΒ gene expression contrary to a previous study undertaken with similar conditions [8]. This may be explained by the fact that, in that work, the stimulation was applied to the Caco-2/TC7 clone but not the traditional Caco-2 cell line, and these data confirm that these two cell types do not quite behave similarly.

The very last part of the study was dedicated to the evaluation of the so-called protective effects of dietary polyphenols on the prevention of the pro-inflammatory response associated with the cocktail of TNF-α, IL-1β, and IFNγ. The pro-inflammatory response on the intestinal epithelium is associated with the loss of its integrity mainly due to the loss of tight junction (TJ) integrity. As such, the scientific community has been looking for the identification of dietary chemo-protectors of the epithelial barrier as an alternative pharmaceutical approach especially when considering nutritional factors in IBD patients [27,28,29]. In this respect, dietary polyphenols have long been studied in an attempt to reduce gut permeability by improving TJ function but also limiting the activation of pro-inflammatory signaling pathways [30]. Several reviews have been dedicated to the evaluation of polyphenols in the management of IBD [31]. Among the many different polyphenols, we opted for the evaluation of flavan-3-ol protective effects. They belong to the most representative classes of dietary polyphenols [32]. Among them, catechins are the most present ones. They are quite abundant in our diet ranging from legumes to fruits such as grapes, litchis, or apples, or cocoa beans and tea leaves for the most concentrated food [10]. Their health benefits in inflammatory bowel diseases have been largely studied (for a review, see [33]). Among the catechin forms available for in vitro studies, we used an antioxidant flavonoid of plant origin, (+)-Cyanidol-3, a catechin hydrate, which preserves its properties at 37 °C and pH = 7 [34]. It also is the main flavan-3-ol monomer in the human diet. When applied on human epithelial cell lines, flavan-3-ols usually promote antioxidant responses by preventing reactive oxygen species production and apoptotic pathway activation [35]. The fully differentiated co-culture of Caco-2/HT29-MTX was incubated with these catechins for 1 h before exposing the epithelium to the cocktail of pro-inflammatory cytokines. None of the parameters studied were modified, in contrast to unstimulated cells in the presence of the polyphenol alone. Furthermore, the catechin was unable to prevent the pro-inflammatory response induced by the cocktail since we did not observe any limitation of IL-8 secretion in the apical medium, regardless of the medium used. These results are similar to a previous study using (+)-catechins on a model of Caco-2 and HT29-MTX co-culture stimulated with a conditioned medium of activated macrophages [14]. In their work, the authors pointed out the fact that (+)-catechins were unable to limit the alteration of gut permeability of the Caco-2/HT29-MTX co-culture and also the prooxidant effect of the medium of activated macrophages [14]. The authors nevertheless pointed out that the polyphenol markedly increased the expression of claudin-7, one of the most common TJ proteins present in the intestinal epithelium. The concentration to which catechins are used is of prime interest since their overdosage may damage the intestinal epithelium as already evidenced in animal models [36]. The literature indeed describes possible adverse effects of polyphenols including pro-oxidant effects, the perturbation of transporters, and the modulation of the activity of a couple of phase I/II enzymes [28]. Interestingly, in this study, the concentration of the catechins was three times higher than the one used here (50 µM vs. 15 µM). As such, the concentration used here could not be considered as too high. Taken altogether, this information rather works in favor of the fact that polyphenols may not act directly on the epithelium to protect it but rather interact, in vivo, with the microbiota to improve intestinal health [37]. The microbiota is able to metabolize polyphenols and generate catabolites promoting health [38]. Furthermore, only 10% of the dietary polyphenols are absorbed in the small intestine. They mainly enter the colon where they are transformed into phenolic acids by the gut microbiota [39]. To confirm this hypothesis, it would then be more appropriate to apply a medium coming from the gut microbiota incubated with the (+) catechins. One cannot exclude, however, that (+)-catechins have difficulties penetrating the cell membrane because of its poor stability and bioavailability due to its high hydrophobicity [40,41]. A previous work evidenced that, by encapsulating this polyphenol, its uptake by a monoculture of Caco-2 cells could be improved [42]. Nevertheless, this conclusion remains to be evaluated in the actual experimental conditions, i.e., a pro-inflammatory stimulus of the co-culture in Transwell^®^ chambers.

Since polyphenols are also considered as potent molecules to induce apoptosis and cell cycle arrest, particularly in the context of prevention of tumorigenesis, we were also interested in evaluating the apoptosis pathways in this study. The regulation of caspase-3 and caspase-9 gene expressions as, respectively, markers of the mitochondrial and intrinsic apoptotic pathways was assessed [43]. In contrast to what was previously evidenced in our lab [8], we were not able to see the upregulation of caspase-3 gene expression in this study. The main difference comes from the use of Caco-2 cells instead of Caco-2/TC7 cells. Furthermore, (+)-catechins did not, by themselves, influence the expression of the two genes, indicating that, under physiological conditions, this pathway is not modulated by this polyphenol.

## 4. Materials and Methods

### 4.1. Cell Culture

The experiments were run using 2 intestinal epithelial cell lines both coming from human colorectal adenocarcinoma: Caco-2 cells and HT29-MTX (European Collection of Authenticated Cell Cultures). The first cell line shows characteristics of enterocytes upon full differentiation after 21 days. These cells were used between passages 56 and 61. The second cell line, by contrast, expresses a goblet cell phenotype and was used between passages 65 to 71. Cells were grown as a co-culture of Caco-2 and HT29-MTX to mimic the intestinal epithelial lining. The two cell lines were routinely grown in an atmosphere of 5% carbon dioxide at 37 °C either in DMEM GlutaMAX (Dulbecco’s Modified Eagle Medium) (Fisher Scientific SAS, Illkirch, France) or RPMI-1640 (Roswell Park Memorial Institute), 1% (*v*/*v*) L-Glutamine (Eurobio Scientific, Les Ulis, France), 1% (*v*/*v*) Sodium pyruvate, and 1% (*v*/*v*) HEPES buffer solution (Dutcher, Issy-les-Moulineaux, France). To both media were added 10% (*v*/*v*) heat-inactivated fetal bovine serum (FBS), 1% (*v*/*v*) non-essential amino acids, and 1% (*v*/*v*) penicillin/streptomycin (Dutcher, Issy-les-Moulineaux, France). Cells were split when they reached an 80% confluency using a trypsin solution (0.25% Trypsin EDTA—Thermofisher Scientific, Illkirch, France).

### 4.2. Co-Culture Stimulation

#### 4.2.1. Time-Dependent Effect of the Growth Medium on the Co-Culture Response to the Pro-Inflammatory Cytokine Cocktail

The co-culture of Caco-2 and HT29-MTX was seeded on a 24-well plate polycarbonate membrane at a density of 1 × 10^5^ cells/well (CytoOne, Starlab, Orsay, France) at a ratio of 9:1 [8] by adding 1 mL of cell suspension and culture medium per well. Cells were grown for 21 days until they reached full differentiation. Two culture media were used: DMEM and RPMI-1640. They were changed every other day during the first two weeks and every day during the last week. On day 20, the cells were rinsed with phosphate-buffered saline (PBS) and put into their respective serum-free medium for 24 h. The cells were then stimulated with a cocktail of pro-inflammatory cytokines: TNF-α (20 ng/mL), IL-1β (1 ng/mL), and IFN-γ (10 ng/mL) (BioTechne, Lille, France) for 2 and 6 h in order to compare the time-dependent response of the co-culture to this pro-inflammatory stimulus. At the end of the stimulation, the media were collected, and the cells harvested in TRIzol (Thermofisher Scientific, Illkirch, France). Samples were frozen at −80 °C until further use.

#### 4.2.2. Incidence of Medium on the Co-Culture Response to Pro- and Anti-Inflammatory Stimuli

The co-culture of Caco-2 and HT29-MTX was seeded on 6-well polycarbonate membrane cell culture inserts with HD (high density) 0.4 µm pores at a density of 4 × 10^5^ cells (Corning, Dutcher, Issy-les-Moulineaux, France) at a ratio of 9:1. The cells were grown for 21 days until they reached full differentiation. Every condition was tested in RPMI-1640 and DMEM to compare the response of the cells to 2 pro-inflammatory stimuli: LPS from *E. coli* and the cocktail of pro-inflammatory cytokines and the protective effect of the catechins (Figure 10). The culture medium was changed every other day during the first two weeks and every day during the last week. The day before stimulation, the cells were rinsed with PBS and put into their respective serum-free medium. Twenty-four hours later, the cells were stimulated with a solution of catechins (Sigma-Aldrich Co, Darmstadt, Germany) (15 µM in ethanol), 1 h before exposing the cells to either the cocktail of the above-mentioned cytokines or a solution of LPS (10 ng/mL) serotype O111:B4 from *E. coli* for 6 h. At the end of the stimulation, the insert medium (called “apical medium”) was harvested and stored at −80 °C until further measurement of markers of intestinal inflammation. The cell layers were harvested in TRIzol and stored at −80 °C before performing RNA extraction.

### 4.3. Measurement of Membrane Permeability Alteration by TEER

In the second series of experiments, Trans-Epithelial Electrical Resistance (TEER) between the apical and the basal medium was measured using an ohmmeter (Millicell, Merck, Darmstadt, Germany) to assess the integrity of the co-culture monolayers before and after the addition of the various stimuli applied. The changes in the TEER were calculated using the following Equation (1).
∆TEER = [(mean of final sample TEER − final control of TEER) × 4.2]-[(mean of initial sample TEER − initial control of TEER) × 4.2].(1)

### 4.4. Real-Time PCR for Gene Expression of Cell Mediators

Total RNA was isolated from cells using TRIzol (Fisher Scientific SAS, Illkirch, France), and RNA concentration was measured spectrophotometrically (NanoDrop 2000, ThermoScientific, Illkirch, France). cDNA was obtained from 5 µg RNA using the Kit GoScript RT (Promega, Charbonnières, France) following the manufacturer’s instructions. Absorbance ratios at 260/280 nm and at 260/230 nm were measured using spectrophotometry to assess the purity of the DNA samples.

Primers and SYBR Green PCR master mix were, respectively, purchased from Eurofins Scientific France (Nantes, France) and Promega (Charbonnières, France) (Table 1). qPCRs were run on a StepOnePlus Real-Time PCR System (Applied Biosystem, Foster City, CA, USA) and the data were processed with the StepOnePlus software v2.3. Reactions were performed in duplicate. The levels of amplified cDNA were calculated using the −∆∆CT method (−∆∆Ct = mean ∆CT_control_ − ∆Ct_exposed_). After testing glyceraldehyde-3-phosphate dehydrogenase (GADPH) and Peptidylprolyl Isomerase A (PPIA), PPIA was kept as a housekeeping gene.

The regulation of the expression of genes coding for the C-X-C motif chemokine ligand 8 (CXCL8), for IL-8, for the p65 subunit of the nuclear factor-kappa B (NF-κΒ), and for Caspase 3 and Caspase 9 was then assessed.

### 4.5. Evaluation of Il-8 Secretion in Media by ELISA

The levels of IL-8 secretion in both the basal and apical media of the co-cultures were measured after the various stimulations. The measurement of IL-8 in the media harvested at the end of stimulation was performed using an ELISA kit according to the manufacturer’s instructions (Human IL-8/CXCL8 DuoSet ELISA, BioTechne, Lille, France).

### 4.6. Statistics

Data were expressed as the mean ± Standard Error of the Mean (SEM) and were analyzed using the GraphPad Prism software (GraphPad Prism version 5 for Windows, GraphPad Software, San Diego, CA, USA). Student’s *t*-test was performed for comparisons of the co-culture sensitivity to pro-inflammatory cytokine stimulation. For all other experiments, a nonparametric Kruskal–Wallis analysis of variance was performed and, when the difference was statistically significant, a Dunn’s post hoc test was then applied to analyze the effects. The threshold for statistical significance was set to *p* < 0.05.

## 5. Conclusions

In conclusion, the co-culture of Caco-2 and HT29-MTX cells in RPMI-1640 medium does not seem to influence the response of the two cell lines to a pro-inflammatory stimulus. It does not change the response to dietary components either, confirming the same level of differentiation of the two cell lines and the suitability of this medium for the settlement of more complexified co-culture models. Furthermore, the response to a cocktail of pro-inflammatory cytokines remains consistent. By contrast, the activation of an inflammatory response by LPS from *E. coli* seems to be more difficult to stimulate not because of a medium effect but rather due to the mucus production by goblet cells. As such, the investigation of pro-inflammatory stimuli and intracellular signaling responses in this fully differentiated co-culture shall be adjusted in duration or intensity to reproduce pathophysiological stimuli observed in IBD. It seems to be important not to neglect the sensitivity of cells to stressful conditions of medium modifications. To avoid variable responses of the culture to LPS from *E. coli*, it is also of importance to thoroughly choose the most appropriate serotype.

However, while the health benefits of polyphenols are widely acknowledged, the demonstration of the protective effect of catechins in vitro is still debatable. This is certainly because of the physicochemical properties of these molecules and more especially their hydrophobicity. The lack of efficiency of catechins in this study cannot be generalized to other polyphenols and shall be tested after a longer period of incubation and other concentrations. Further research shall be conducted to evaluate the catabolism of polyphenols and their benefits on health [44].

This work nevertheless contributed to a better understanding of the complexity of in vitro cellular response. It also pointed out the need of pursuing the investigation on the characterization of cellular response. Furthermore, while being widely used by the scientific community, this co-culture model is of carcinogenic origin with different cellular response. It would be necessary to move onto non-carcinogenic human cell lines to better simulate what is observed in the human intestinal epithelium. Lastly, although several attempts have been made to complexify the actual in vitro models, none are yet able to reproduce the complex crosstalk existing between the gut microbiota, the epithelium, and the immune system to regulate and maintain intestinal homeostasis. More work remains to be carried out, in line with the one presented here, to move onto a satisfactory in vitro model to evaluate the alterations observed in IBD patients.

## Figures and Tables

**Figure 1 ijms-25-07360-f001:**
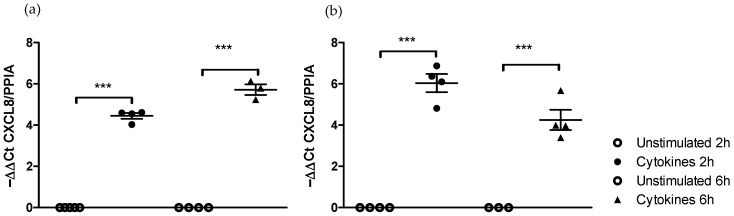
Time-dependent regulation of CXCL8 gene expression after incubation with a cocktail of pro-inflammatory cytokines in DMEM and RPMI-1640 media. (**a**) CXCL8 expression in DMEM. (**b**) CXCL8 expression in RPMI-1640. *** Significantly different (*p* < 0.001) from unstimulated cells.

**Figure 2 ijms-25-07360-f002:**
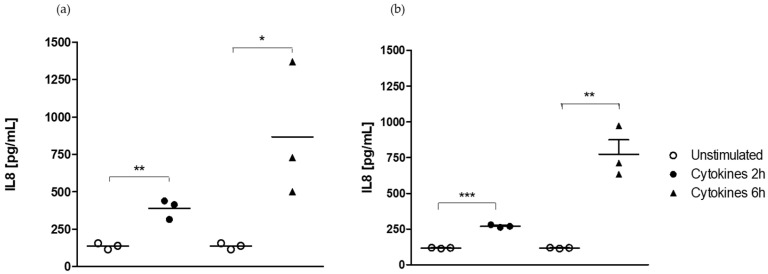
Time-dependent regulation of IL-8 secretion after incubation with a cocktail of pro-inflammatory cytokines in DMEM and RPMI-1460 media. (**a**) Il-8 secretion in DMEM. (**b**) Il-8 secretion in RPMI-1640. *, **, *** Significantly different (*p* < 0.05; *p* < 0.01; *p* < 0.001) from unstimulated cells.

**Figure 3 ijms-25-07360-f003:**
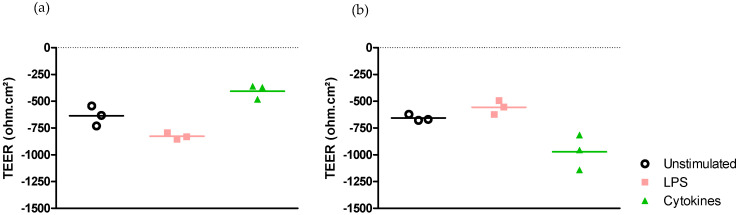
Effect of medium on permeability modulation after stimulation with LPS from *E. coli* or a cocktail of pro-inflammatory cytokines. (**a**) Alteration of permeability in DMEM medium. (**b**) Alteration of permeability in RPMI-1640 medium.

**Figure 4 ijms-25-07360-f004:**
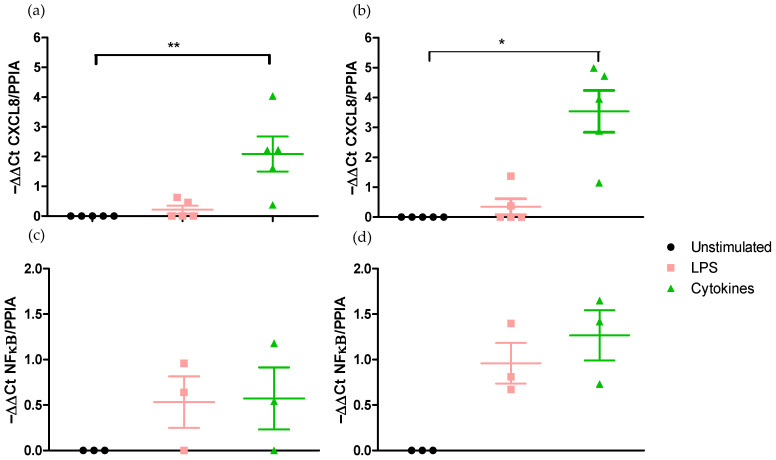
Effect of medium on the modulation of pro-inflammatory NK-κΒ and CXCL8 gene expression after 6 h exposure to LPS or a cocktail of pro-inflammatory cytokines. (**a**) Modulation of CXCL8 expression in DMEM medium. (**b**) Modulation of CXCL8 expression in RPMI-1640 medium. (**c**) Modulation of NF-κΒ expression in DMEM medium. (**d**) Modulation of NF-κΒ expression in RPMI-1640 medium. *, ** Significantly different (*p* < 0.05; *p* < 0.01) from unstimulated cells.

**Figure 5 ijms-25-07360-f005:**
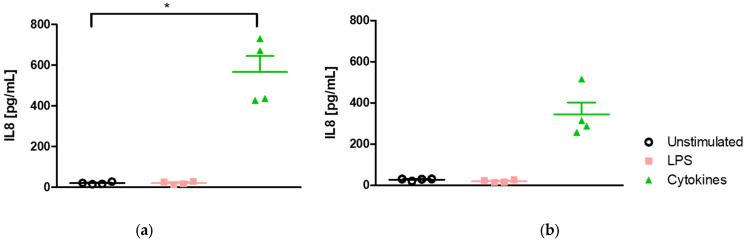
Effect of medium on IL-8 secretion in the apical medium after 6 h of exposure with LPS or the cocktail of pro-inflammatory cytokines (**a**) Secretion of IL-8 in DMEM medium. (**b**) Secretion of Il-8 in RPMI-1640 medium * Significantly different (*p* < 0.05) from unstimulated cells.

**Figure 6 ijms-25-07360-f006:**
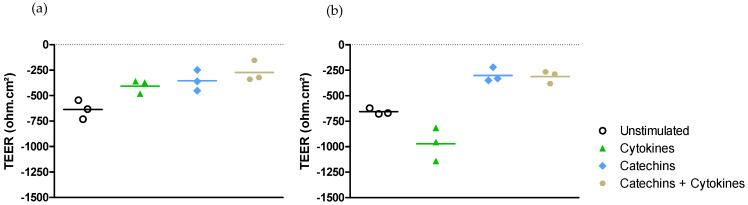
Effect of medium on transepithelial permeability modulation by catechins after stimulation with a cocktail of pro-inflammatory cytokines. (**a**) Permeability modulation in DMEM medium. (**b**) Permeability modulation in RPMI-1640 medium.

**Figure 7 ijms-25-07360-f007:**
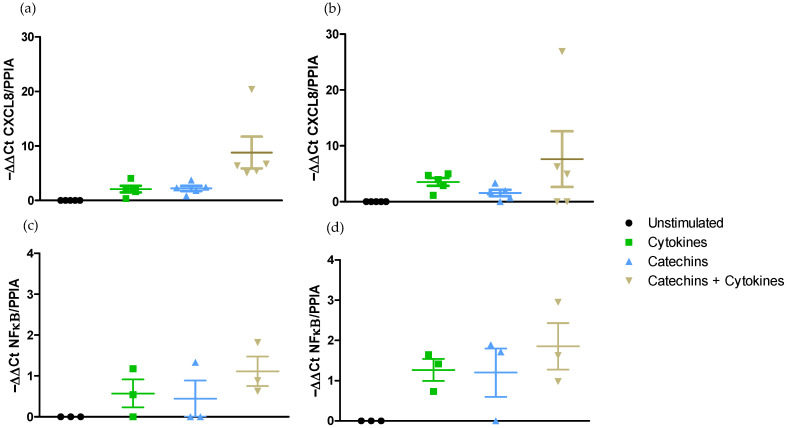
Effect of medium on NF-κΒ and CXCL8 expression modulation by catechins after stimulation with a cocktail of pro-inflammatory cytokines. (**a**) CXCL8 gene expression in DMEM medium. (**b**) CXCL8 gene expression in RPMI-1640 medium. (**c**) NF-κΒ gene expression in DMEM medium. (**d**) NF-κΒ gene expression in RPMI-1640 medium.

**Figure 8 ijms-25-07360-f008:**
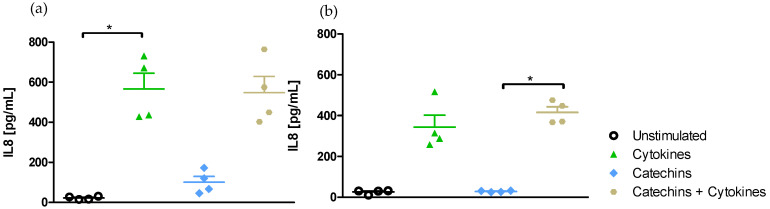
Effect of medium on IL-8 secretion modulation after exposure to catechins and a cocktail of pro-inflammatory cytokines. (**a**) IL-8 secretion modulation in DMEM medium. (**b**) IL-8 secretion modulation in RPMI-1640 medium. * Significantly different (*p* < 0.05; *p* < 0.01) from unstimulated cells. + Significantly different (*p* < 0.05) from catechin-stimulated cells.

**Figure 9 ijms-25-07360-f009:**
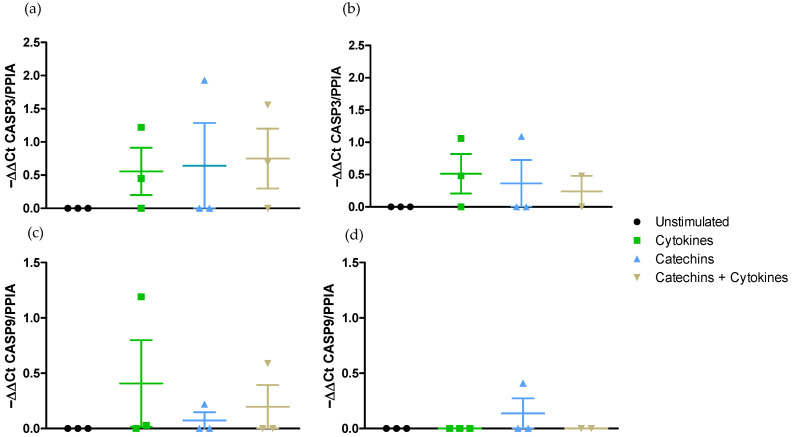
Effect of medium on CASP3 and CASP9 expression modulation by catechins after stimulation with a cocktail of pro-inflammatory cytokines. (**a**) CASP3 gene expression in DMEM medium. (**b**) CASP3 gene expression in RPMI-1640 medium. (**c**) CASP9 gene expression in DMEM medium. (**d**) CASP9 gene expression in RPMI-1640 medium.

**Figure 10 ijms-25-07360-f010:**
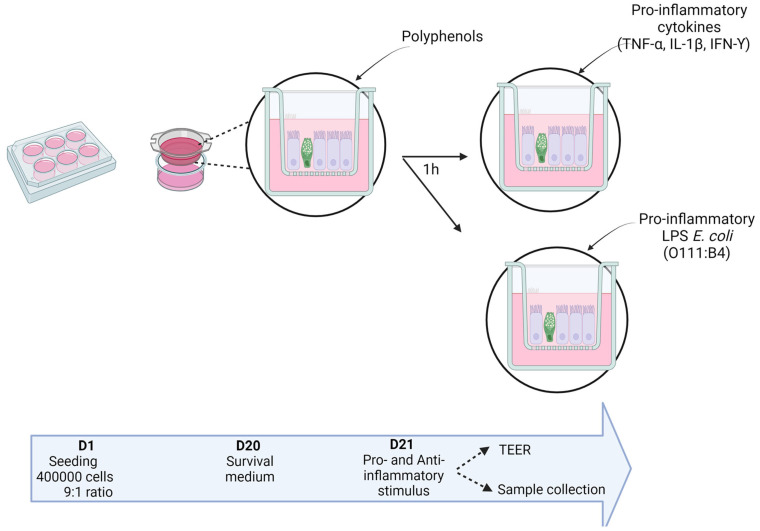
Experimental procedure of the pro- and anti-inflammatory stimulation in the Caco-2/HT29-MTX co-culture materials and methods used (personal creation elaborated with Biorender.com).

**Table 1 ijms-25-07360-t001:** Primers used in the project.

Function	Gene	Name	Sequences 5′-> 3′ or Reference	Su
Housekeeping Gene	PPIA	Peptidylprolyl Isomerase A	CCTATCCTAGAGGTGGCGGA TCATCGCAGAAGGAACCAGAC	Eurofins
Inflammatory Genes	CXCL8	Interleukin 8	AGAGTGATTGAGAGTGGACC ACTTCTCCACAACCCTCTG	Eurofins
	NF-κB	Nuclear Factor κB p65 subunit	GGGGGCATCAAACCTGAAGA GGAGAGAAGTCCCCAAAGGC	Eurofins
Apoptosis Genes	CASP3	Caspase 3	QT00029162	QIAGEN
	CASP9	Caspase 9	QT00036267	QIAGEN

## Data Availability

The datasets presented in this article are not readily available because the data are part of an ongoing study. Requests to access the datasets should be directed to the corresponding author.

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
