# Peer review of "Intestinal Epithelial Co-Culture Sensitivity to Pro-Inflammatory Stimuli and Polyphenols Is Medium-Independent"

_ijms, 2024, doi:10.3390/ijms25137360_

Round 1
Reviewer 1 Report
Comments and Suggestions for Authors
The paper of Haddad et al is a very interesting paper about an in vitro intestinal epithelium model exposed to pro-inflammatory stimulation as well as dietary polyphenols. The paper is well-written, the conclusions are supported by the scientific data and the discussion section is extensive.
The authors used relevant references for the research.
Author Response
We thank the reviewer for his/her feedback.
Reviewer 2 Report
Comments and Suggestions for Authors
The authors’ study offers a better understanding of the complexity of in vitro cellular response and points out the need of the investigation on the characterization of cellular response.
Scientific research confirmed that dietary polyphenols possess both protective and therapeutic effects in the management of IBD mediated via down-regulation of inflammatory cytokines and enzymes, enhancing antioxidant defense, and suppressing inflammatory pathways and their cellular signaling mechanisms10.2174/1389201016666150118131704
Polyphenol intake improves the health effects of the gastrointestinal microbiota by activating the intestinal immune function, SCFA (short-chain fatty acids) excretion, and other physiological processes10.3390/molecules24020370.
The total intake of dietary polyphenol at the small intestine is estimated at around 10% 10.1093/jn/130.8.2073S . Hence, a large proportion of ingested polyphenols are transported to the large intestine, where they are catabolized to phenolic acids by the intestinal bacteria. Recently, much attention has been paid to the bioavailability and physiological actions of their catabolites and to the functional interaction between polyphenols and intestinal bacteria.
The intake of polyphenols improves the health effects of the intestinal microbiota by activating SCFA excretion, intestinal immune function, and other physiological processes. The microbiota-dependent effects of polyphenols may to be applied practically to the health food or supplement industries. For this purpose, further research is necessary to analyze the catabolic reactions of polyphenols and their reaction products and to determine the mechanisms of action of these compounds on the intestinal microbiota. 10.1126/science.aau5812; 10.3389/fmicb.2017.02171
The author's study has a comprehensive design, the results provide a better understanding of the cellular immune response and the role of dietary and microbial metabolites in the regulation of host immunity.
The authors should focus in the Discussion chapter on the role of dietary and microbial metabolites in the regulation of host immunity, adding the suggested references.
Comments on the Quality of English LanguageModerate English Editing is required
Author Response
We thank the reviewer for his/her comments. We have read with attention the 5 suggested references brought to our attention. We would like to point out that the experiments run in the article are focused on intestinal epithelial cells but not colonic cells. Furthermore, the microbiota is not present in this model. The polyphenols, which are plant secondary metabolites were not metabolized. We only administered the exotoxin i.e. LPS but not the whole bacteria: this approach does not lead to any bacterial metabolism. Although the scope of the article was not fully dedicated to the understanding of the role of dietary and microbial metabolites, we added the above cited references in the discussion and the conclusion of the paper.
Reviewer 3 Report
Comments and Suggestions for Authors
The manuscript entitled "Intestinal epithelial co-culture sensitivity to pro-inflammatory stimuli and polyphenols is medium independent." aimed to compare the sensitivity of epithelial cells to proinflammatory stimuli and also to dietary polyphenols in 14 both DMEM and RPMI-1640 media.
The abstract is not written following standard scientific guidelines for structuring abstracts.
The first impression that the manuscript leaves is a need for extensive English language, scientific style, and grammatical revision.
The main issue of the work is a minor contribution to the scientific field.
No need for a full-stop after the title.
Kindly explain what you mean by "pathogenic molecules".
line 47 "Because they express those tight interactions between the microbiota" - not understandable
line 49 "Nevertheless, mainly for ethical reasons, the understanding of the chronology of the induction of the inflammatory response and barrier disruption at the intestinal level cannot be assessed by such a mean" - not understandable
line 53 - kindly explain 2D models vs. 3D
line 63 ". Since these cells are very sensitive to the growth medium, it becomes evident 63 that their use for studying their interactions with the intestinal epithelium will necessitate 64 an adaptation of the growth medium of the other cell lines to limit the risk of altering the 65 cell response." - kindly rewrite the sentence to make it understandable and logical
line 88 "As thus, since to our knowledge, so far," - not correctly written, kindly rewrite
No need to rewrite the methodology in the discussion section.
Comments on the Quality of English Language
English language needs major revision.
Author Response
Comments and Suggestions for Authors : The manuscript entitled "Intestinal epithelial co-culture sensitivity to pro-inflammatory stimuli and polyphenols is medium independent." aimed to compare the sensitivity of epithelial cells to proinflammatory stimuli and also to dietary polyphenols in 14 both DMEM and RPMI-1640 media.
The abstract is not written following standard scientific guidelines for structuring abstracts.
We do not quite understand the comment of the reviewer since the structure of the abstract is following the template of the journal.
The first impression that the manuscript leaves is a need for extensive English language, scientific style, and grammatical revision.
The overall manuscript has been thoroughly revised.
The main issue of the work is a minor contribution to the scientific field.
- No need for a full-stop after the title.
We thank the reviewer for the comment and for taking the time to assess our manuscript. The full stop was removed after the title.
- Kindly explain what you mean by "pathogenic molecules".
Pathogenic molecules are elements that may induce an inflammatory reaction from living cells or organisms. They could be xenobiotics or neoformed compounds that may disrupt the intestinal barrier by promoting inflammation and tissue damage. This information has been added in the manuscript (line 31).
- line 47 "Because they express those tight interactions between the microbiota" - not understandable
We meant to say that the animal models are an accurate representation of human physiology, since they express the underlying cross talk between the intestinal epithelium, the microbiota and the immune system. To ease the understanding of the reader, we have modified the sentence and hope that now it is clearer (line 47).
- line 49 "Nevertheless, mainly for ethical reasons, the understanding of the chronology of the induction of the inflammatory response and barrier disruption at the intestinal level cannot be assessed by such a mean" - not understandable
The above-mentioned sentence relates to animal models. Their use shall be restricted to the evaluation of physiological processes that cannot be assessed otherwise. When we want to study the kinetics/chronology of the induction of the inflammatory response, the use of cellular models is more convenient and ethically appropriate (to limit the use of animals). Furthermore, because of differences in the responses between species and sometimes the difficulty of extrapolating results to humans, the need to find co-culture models representing human physiology in vivo is crucial.
To ease the understanding of the sentence, we have replaced “by such a mean” by “with animal models” (line 51).
- line 53 - kindly explain 2D models vs. 3D
In 2D cell culture models, cells are grown on a flat surface whereas in 3D models, cells are grown in a three-dimensional space embedded in a gel-like matrix to mimic the extracellular matrix and therefore mimic better the tissue microenvironment. This information has been added in the manuscript to improve the understanding (lines 54 & 56).
- line 63 ". Since these cells are very sensitive to the growth medium, it becomes evident 63 that their use for studying their interactions with the intestinal epithelium will necessitate 64 an adaptation of the growth medium of the other cell lines to limit the risk of altering the 65 cell response." - kindly rewrite the sentence to make it understandable and logical
To ease the understanding, we have replaced the sentence by the following one. “Literature has shown that THP-1 can only grow in RPMI 1640 medium. Therefore, to create a functional model reproducing the intestinal epithelium, the growth medium of the other cell components needs to be adjusted to prevent the alteration of the cellular response” (lines 68 to 71).
- line 88 "As thus, since to our knowledge, so far," - not correctly written, kindly rewrite
We have amended the sentence with “According to literature, no study has tackled the response of [….]” (line 94).
- No need to rewrite the methodology in the discussion section.
We wanted to add a small recap of the experiments done to help the reader stay in context of the goal of this study. As per suggested by the reviewer, we have adjusted it.
Comments on the Quality of English Language : English language needs major revision.
The overall manuscript has been fully revised.
Reviewer 4 Report
Comments and Suggestions for Authors Haddad et all attempted to evaluate Caco2/HT29 intestinal cell model in RPMI medium suitable for maintaining immune cell cultures. While the results are valid, and groups working with these models will find them interesting and useful in their work, the organization and presentation of the manuscript is critically lacking. Major concerns: Change colors used in the figures, the chosen greens are not visible and do not contrast well with each other. Error bars on figures must have vertical lines on both sides so it is clear what is being compared. Must keep the same style of graphing and presenting results, each figure 1, 2, and 3 uses a different style which is not acceptable. 132-136: Confusing and unclear language, edit. 130-240: The presentation of results is extremely unclear due to lacking of proper English and very little organization. This section must be edited and re-organized for clarity of the presentation. 276: 10 ng/ml LPS is too low, many studies use up to 100-1000 ng/ml to ensure induction. The discussion is unnecessary too long and convoluted. Figure 10 is unnecessary. Conclusions are unnecessary long. Minor concerns: 39: Change to “leading to” 47-48: Confusing language, please rephrase 77: Change “large bunch” to “large number” 116: Add units to secreted amounts 123: Add units to secreted amounts 129: Change to “but not on the medium” 132, 143: Remove black don’t as the list format is not used in scientific publicationsComments on the Quality of English Language
Moderate editing of English language required
Author Response
Comments and Suggestions for Authors :
Haddad et all attempted to evaluate Caco2/HT29 intestinal cell model in RPMI medium suitable for maintaining immune cell cultures. While the results are valid, and groups working with these models will find them interesting and useful in their work, the organization and presentation of the manuscript is critically lacking.
Major concerns:
Change colors used in the figures, the chosen greens are not visible and do not contrast well with each other.
We thank the reviewer for his/her valuable comments. To ease the reading of figures, green has been removed and replaced by filled and empty signs. We hope that this new presentation is easier to follow.
Error bars on figures must have vertical lines on both sides so it is clear what is being compared.
Error bars on figures have been implemented with vertical lines on both sides to ease the understanding of comparison.
Must keep the same style of graphing and presenting results, each figure 1, 2, and 3 uses a different style which is not acceptable.
All figures have been checked and amended to look alike. We hope that this new presentation allows a better understanding of the results.
132-136: Confusing and unclear language, edit.
The text has been amended
130-240: The presentation of results is extremely unclear due to lacking of proper English and very little organization. This section must be edited and re-organized for clarity of the presentation.
The overall results section has been amended for clarity
276: 10 ng/ml LPS is too low, many studies use up to 100-1000 ng/ml to ensure induction.
We do agree with the reviewer on the fact that most of the studies run with LPS used high concentrations of the exotoxin. However, as explained in the discussion, the concentration to be used greatly depends on the serotype used, and the concentration and serotype used here relate to the following reference doi:10.1016/j.ajpath.2012.10.014.
The discussion is unnecessary too long and convoluted.
The discussion has been constructed according to the instructions of authors.
Figure 10 is unnecessary.
We drew figure 10 to ease the understanding of the experimental procedure at a glance.
Conclusions are unnecessary long.
The conclusions follow the instructions of the journal. However, if necessary, we could summarize them.
Minor concerns:
39: Change to “leading to”
The modification was done.
47-48: Confusing language, please rephrase:
The sentence has been amended to ease the understanding. In fact, the comma after microbiota has been removed to understand that the close interactions relate to the cross communication between the microbiota the immune system and the epithelium.
77: Change “large bunch” to “large number”
The modification was done
116: Add units to secreted amounts
The modification was done
123: Add units to secreted amounts
The modification was done
129: Change to “but not on the medium”
The modification was done
132, 143: Remove black don’t as the list format is not used in scientific publications
As per suggested in the journal instructions, bullet points are recommended to separate information after sub-headings X.Y.Z. We just have followed the template. However, we have added an empty space to better separate the paragraphs.
Comments on the Quality of English Language : Moderate editing of English language required
The English editing was carefully checked, and we hope this version of the manuscript is more pleasant to read.
Round 2
Reviewer 3 Report
Comments and Suggestions for Authors
Although authors have made some changes, the main concern remains low scientific impact, non-scientific way of expression, and a plethora of misleadings.
For example, following the authors' explanation, the question of the meaning of "pathogenic molecules" shows the inadequateness of the terminology used throughout the manuscript.
The abstract is not written in a standard scientific way.
Figures are almost unreadable.
Comments on the Quality of English LanguageEnglish language still needs extensive revision.
Author Response
Although authors have made some changes, the main concern remains low scientific impact, non-scientific way of expression, and a plethora of misleadings. For example, following the authors' explanation, the question of the meaning of "pathogenic molecules" shows the inadequateness of the terminology used throughout the manuscript. The abstract is not written in a standard scientific way. Figures are almost unreadable. English language still needs extensive revision.
We have read with attention the comments of the reviewer but found it hard to improve the overall manuscript due to general consideration that do not help to revise it. To ease the understanding and avoid as much as possible misleading the phrase ” pathogenic molecules” has been replaced by “noxious substances” even though pathogenic molecules include bacterial LPS, one of the substances used in the study. English editing has, once again, been checked but we do not get what a more scientific way of expression is. It would have been interesting to have more explanation by the reviewer on this point. The figures had been amended on demand of the reviewer to all appear the same as thus we do not get why they appear unreadable in this version. We would be greatful if the reviewer could develop especially regarding the scientific way of expression.